# Learning when to trust distant supervision: An application to low-resource POS tagging using cross-lingual projection

## Abstract

Cross lingual projection of linguistic annotation suffers from many sources of bias and noise, leading to unreliable annotations that cannot be used directly. In this paper, we introduce a novel approach to sequence tagging that learns to correct the errors from cross-lingual projection using an explicit noise layer. This is framed as joint learning over two corpora, one tagged with gold standard and the other with projected tags. We evaluated with only 1000 tokens tagged with gold standard tags, along with more plentiful parallel data. Our system equals or exceeds the state-of-the-art on eight simulated low-resource settings, as well as two real low-resource languages, Malagasy and Kinyarwanda.

## 1 Introduction

Part-of-speech tagging is a critical task for natural language processing (NLP) applications, providing lexical syntactic information. Automatic POS tagging has been wildly successful on many rich resource languages using supervised learning over large training corpora (McCallum et al., 2000; Lafferty et al., 2001; Ammar et al., 2016). However, learning POS taggers for low-resource languages from small amounts of annotated data is very challenging (Garrette and Baldridge, 2013; Duong et al., 2014a). For such problems, distant supervision via heuristic methods can provide cheap but inaccurately labelled data (Mintz et al., 2009; Takamatsu et al., 2012; Ritter et al., 2013; Plank et al., 2014). A compromise, considered here, is to use a mixture of both resources: a small collection of clean annotated data and noisy "distant" data.

A popular method for distant supervision is to use parallel data between a low-resource language and a rich-resource language. Although annotated data in low-resource languages are difficult to obtain, bilingual resources are more plentiful. For example parallel translations into English are often available, in the form of news reports, novels or the Bible. Parallel data allows annotation from the high-resource language to be projected across alignments to the low-resource language, which has been shown to be effective for several language processing tasks including POS tagging (Yarowsky et al., 2001; Das and Petrov, 2011; Duong et al., 2013), named entity recognition (Wang and Manning, 2013) and dependency parsing (McDonald et al., 2013).

Although cross-lingual POS projection is popular it has several problems, including noise from poor word alignments (Täckström et al., 2013; Das and Petrov, 2011) and cross-lingual syntactic divergence (Duong et al., 2013). Previous work has proposed heuristics or constraints to clean the projected tag before or during learning. In contrast, we consider compensating for these problems explicitly, by learning a noise transformation to encode the mapping between 'clean' tags and the kinds of noisy tags produced from projection.

We propose a new neural network model for sequence tagging in a low-resource language, suitable for training with both a tiny gold standard annotated corpus, as well as distant supervision using cross lingual tag projection. Our model uses a bidirectional Long Short-Term Memory (BiLSTM), which produces two types of output: gold tags generated directly from the hidden state of neural network, and uncertain projected tags generated after applying a further linear transformation. This transformation, which we refer to as *output noising* encodes the mapping between the projected high-resource tags and low-resource

tags, and learning when and how much to trust the projected data. For example, for languages without determiners, the model can learn to map projected determiner tags to nouns, or if verbs are often poorly aligned, the model can learn to effectively ignore the projected verb tag, through associating all tags with verbs. Our model is trained jointly on gold and distant projected annotations, and can be trained end-to-end with backpropagation.

Our approach captures the relations among tokens, noisy projected POS tags and ground truth POS tags. Our work differs in the use of projection, in that we explicitly model the transformation between tagsets as part of a more expressive deep learning neural network. Our contributions fall in three aspects. First, we study the noise of projected data in word alignments and describe it with an additional layer model. Second, we integrate the model into a deep neural network and jointly train the model on both annotated and projected data to make the model learn from better supervisions. Finally, evaluating on eight simulated and two real-world low-resource languages, experimental results demonstrate that our approach uniformly equals or exceeds existing methods on simulated languages, and achieves 86.3% accuracy for Malagasy and 82.5% on Kinyarwanda, exceeding the state-of-the-art results (Duong et al., 2014a).

## 2   Related Work

For most natural language processing tasks, the conventional approach to developing a system is to use supervised learning algorithms trained on a set of annotated data. However, this approach is inappropriate to low-resource languages due to the lack of annotated data. An alternative approach is to harness different source of information aside from simple annotated text. Knowledge-bases such as dictionaries are one possible source of information, which can be used to to inform or constrain models, such as limiting the search space for POS tagging (Banko and Moore, 2004; Goldberg et al., 2008; Li et al., 2012).

Parallel bilingual corpora provide another important source of information, and are often plentiful even for many low-resource languages in the form of multilingual government documents, book translations, multilingual websites, etc. Word alignments can provide a bridge to project information from a resource-rich source language to a resource-poor target language. For example, parallel data has been used for named entity recognition (Wang and Manning, 2013) based on the observation that named entities are most often preserved in translation; and also in syntactic tasks such as POS tagging (Yarowsky et al., 2001; Das and Petrov, 2011; Duong et al., 2013) and dependency parsing (McDonald et al., 2013). Clues from related languages can also compensate for the lack of annotated data, as we expect there to be information shared between closely related languages in terms of the lexical items, morphology and syntactic structure. Some successful applications using language relatedness information are dependency parsing (McDonald et al., 2011), where a parser is estimated from a source, resource-rich language, but then applied to a target, low-resource language, and POS tagging (Hana et al., 2004) where parts of the tagger are estimated from the source language. However, these approaches are limited to closely related languages such as Czech and Russian, or Telugu and Kannada, and it is unclear whether these techniques will work well in situations where parallel data only exists for less-related languages, as is often the case in practice.

To summarize, for all these mentioned tasks, lexical resources are valuable sources of knowledge, but are also costly to build. Language relatedness information is applicable for closely related languages, but it is often the case that a given low-resource language does not have a closely-related, resource-rich language. Parallel data therefore appears to be the most realistic additional source of information for developing NLP systems for low-resource languages (Yarowsky et al., 2001; Duong et al., 2014b; Guo et al., 2016; Guo et al., 2015), and here we primarily investigate methods to exploit parallel texts.

Yarowsky et al. (2001) pioneered the use of parallel data for projecting POS tag information from a resource-rich language to a resource-poor language. Duong et al. (2014b) proposed an approach using a maximum entropy classifier trained on 1000 tagged tokens, and used projected tags as auxiliary outputs. Das and Petrov (2011) used parallel data and exploited graph-based label propagation to expand the coverage of labelled tokens. Our work is closest to Duong et al. (2014a), and we share the same evaluation setting, which we

believe is well suited to the low-resource applications. Our approach differs from theirs in two ways: first we propose a deep learning model based on a long short term memory recurrent structure versus their maximum entropy classifier, and secondly we model the projection tag explicitly as a form of noise applied after classification, while they attempt to capture the correlations between tagsets only implicitly through a joint feature set over both tags. We believe that our work is the first to explicitly model the noise affecting cross-lingual projected annotations, and thereby allowing this rich data resource to be better exploited in learning NLP models in low-resource languages.

## 3 Framework

In this work, we consider the POS tagging problem for a low-resource language using both the gold annotated data and distant projected data. For a low-resource language, we assume two sets of data. First, there is a small conventional corpus for the low-resource language, annotated with gold tags. Second, there is also a parallel corpus between the language and English, where we can reliably tag the English side and project these annotations across the word alignments. Then based on the annotated and the projected data, we learn a deep neural model for the POS tagging. The goal of learning here is to improve the POS tagging accuracy on the low resource language.

### 3.1 POS projection via word alignments

Parallel data is often available for low-resource languages. For example, for Malagasy we can obtain bilingual documents with English directly from the web. This provides ample opportunity for projecting annotations from English into the low-resource language. Although the POS tags can be projected, given sentence and word-alignments, direct projection has several issues and results in very noisy and unreliable annotations (Yarowsky et al., 2001; Duong et al., 2014b). One source of error are the word alignments. These errors arise from words in the source language that are not aligned any words in the other language, which might be due to their not being translated closely, errors in alignments, or translation phenomena that do not fit the assumptions underlying the word based alignment models (e.g., many to many translations cannot be captured).

**Word alignments:**

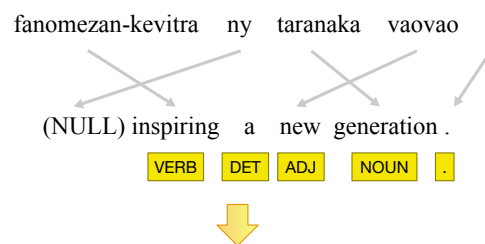

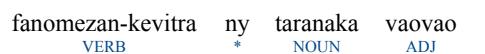

**Projection:**

Figure 1: An example of POS projection via word alignments. * indicates unknown POS tag, which we treat as having tag distribution over all tokens in the source sentence (in the example, a uniform mix of VERB, DET, ADJ, NOUN and '.'.)

An example of POS projection via word alignments between Malagasy and English is shown in Figure 1. A word in Malagasy is connected to a word in English or NULL word. Thus there exist words in the target language which are not aligned a word from source, for example `ny` in Figure 1. Previous works either used the majority projected POS tag for a token or used a default value to represent the token (Duong et al., 2014a; Täckström et al., 2013). Another problem is about noisy projected tags. For example, in this sentence, *fanomezan-kevitra* is labelled as VERB incorrectly, but should be NOUN, a consequence of a non-literal translation.

We now turn to the manner of labelling the projected data. For the parallel data, we consider each token in the low-resource language. Where this token is aligned to a single token in English, we assign the tag for that English token. For tokens that are aligned to many English words or none at all (NULL), we assign a distribution over tags according to the tag frequency distribution over the whole English sentence.

A natural question is whether this projected labelling might be suitable for use directly in supervised learning of a POS tagger. To test this, we compare training a bidirectional Long Short-Term Memory (BiLSTM) tagger on this data, a small 1000 token dataset with gold-standard tags, and the union of the two.[1] Evaluating the tag-

---

[1]See §3.2 for the model details, and §4.1 for a description

ging accuracy against gold standard tags, we observe in Tables 1 and 2 (top section, rows labelled BiLSTM) that the use of the gold-standard (Annotated) data is considerably superior to training on the directly Projected data, despite the smaller amount of Annotated data, while using the union of the two datasets does result in mild improvements in a few languages, but worsens performance for others.

These sobering results raise the question of how we might use the bilingual resources in a more effectively manner than direct projection. Clearly projections contain useful information, as the tagging accuracy is well above chance, however they are riddled with noise and biases, which needs to be accounted for for adequate performance.

### 3.2 BiLSTM with noise layer

To address this problem, we propose a model based on jointly modelling the clean annotated data and the noisy projected data. For this we use a bidirectional LSTM tagger, as illustrated on the left in Figure 2, although other classifiers could be easily used in its place. The BiLSTM offers access to both the left and right lexical contexts around a given word (Graves et al., 2013), which are likely be of considerable use in POS tagging where context of central importance.

Let $x_t$ indicate a word in a sentence and $y_t$ indicate its corresponding POS tag, where $K$ is the size of the tagset.[2] The recurrent layer is designed to store contextual information, while the values in the hidden and output layers are computed as follows:

$$\overrightarrow{h}_t = \mathtt{lstm}(\overrightarrow{h}_{t-1}, x_t)$$
$$\overleftarrow{h}_t = \mathtt{lstm}(\overleftarrow{h}_{t+1}, x_t)$$
$$o_t = \mathrm{softmax}(W_{\rightarrow}\overrightarrow{h}_t + W_{\leftarrow}\overleftarrow{h}_t + b) \quad (1)$$
$$y_t \sim \mathrm{Multinomial}(o_t).$$

This supervised model is trained on annotated gold data in the standard manner using a cross-entropy objective with stochastic gradient descent through the use of gradient backpropagation.

The projected data, however, needs to be treated differently to the annotated data: the tagging is often uncertain, as tokens may have been aligned to words with different parts of speech, be multiply

aligned or left unaligned. These tags are not to be trusted in the same way as the gold annotated data. Our work allows for noise explicitly in the training objective, by attempting to model the noise generating process. The projected data consists of pairs, $(x_t, \tilde{y})$, where $\tilde{y}$ denotes the projected POS tag. In this setting, we assume that the true label, $y_t$, is latent variable and both $\tilde{y}$ and $y$ are $K$-dimensional binary random variables: $\tilde{y}_t$ is a vector representation of a projected tag, and $y_t$ is a one-hot representation of a gold tag.

We augment the deep neural network model to include a noise transformation such that its prediction matches the POS tag distribution of the noisy data, as follows:

$$p(\tilde{Y}_t = j | x_t, \theta, A) = \mathrm{softmax}\left(\sum_i a_{i,j} o_{t,i}\right), \quad (2)$$

where $o_{t,i} = p(Y_t = i | x_t, \theta)$ is the probability of tag $i$ in position $t$ according to (1). This equation is parameterized by a $K \times K$ matrix $A$.[3] Each cell $a_{i,j}$ denotes the confusion score between classes $i$ and $j$, with negative values quashing the correspondance, and positive values rewarding a pairing; in the situations where the projected tags closely match the supervised tagging, we expect that $A \propto I$.

Joint modelling of the gold supervision and projected data gives rise to a training objective combining two cross-entropy terms,

$$\mathcal{L}(\theta, A) = -\frac{1}{|T^p|} \sum_{t \in T^p} \langle \tilde{y}_t, \log \mathrm{softmax}(A o_t) \rangle$$
$$-\frac{1}{|T^t|} \sum_{t \in T^t} \langle y_t, \log o_t \rangle,$$

where $T^p$ indexes all the token positions in the projected dataset, and similarly for $T^t$ over the annotated training set.

We illustrate the combined model in Figure 2, showing on the left the gold supervised model and on the right the distant supervised components. The distant model builds on the base part, through feeding the output through a noising layer, which is finally used in a softmax to produce the noised output layer. The matrix $A$ parameterizes the final layer, to adjust the tag probabilities from

---

of the datasets and evaluation.

[2] We use the universal tagset from (Petrov et al., 2011), enabling comparison across languages.

[3] Our approach also supports mismatching tagsets, in which case $A$ would be rectangular with dimensions based on the sizes of the two tag sets.

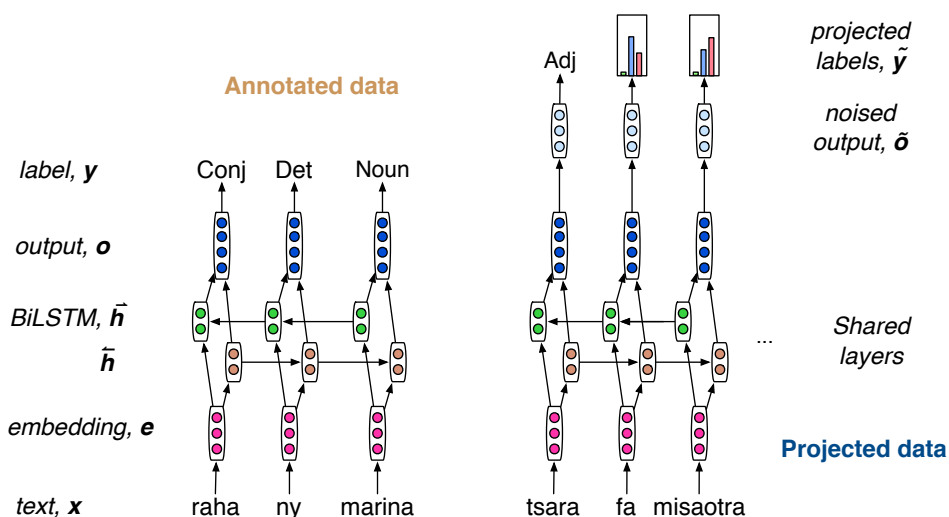

Figure 2: Illustration of the model architecture, using a bidirectional LSTM recurrent network, with a tag classification output. The left part illustrates the supervised training scenario and test setting, where each word $x$ is assigned a tag $y$; the right part shows the projection training setting, with a noise layer, where the supervision is either a projected label or label distribution (used for NULL aligned words).

the supervised model into a distribution that better matches the noisy projected POS tags. However, the ultimate goal is to predict the POS tag $y_t$. Consider the training effect of the projected POS tags: when performing error backpropagation, the cross-entropy error signal must pass through the tag transformation linking $\tilde{o}$ with $o$, which can be seen as a de-noising step, after which the cleaned error signal can be further backpropagated to the rest of the model. Provided there are consistent patterns of noise in the projection output, this technique can readily model these sources of variation, with a tiny handful of parameters, and thus greatly improve the utility of this form of distant supervision.

Directly training the whole deep neural network with random initialization is impractical, because without a good estimate for the $A$ matrix, the noise from the projected tags may misdirect training result in a poor local optima. For this reason the training process contains two stages. On the first stage, we use the clean annotated data to pretrain the network. On the second stage, we jointly use both projected and annotated data to continue training the model.

## 4 Experiments

We evaluate our algorithm on two kinds of experiment settings, simulation experiments and real-world experiments. For the simulation exper-

iments, we use the following 8 European languages: Danish (da), Dutch (nl), German (de), Greek (el), Italian (it), Portuguese (pt), Spanish (es), Swedish (sv). These eight languages are obviously not low-resource languages, however we can use this data to simulate the low-resource setting by only using a small fraction of the gold annotations for training. This evaluation technique is widely used in previous work, and allows us to compare our results with prior state-of-the-art algorithms. For the real-world experiments, we use the following two low-resource languages: Malagasy, an Austronesian language spoken in Madagascar, and Kinyarwanda, a Niger-Congo language spoken in Rwanda.

### 4.1 Evaluation Corpora

#### 4.1.1 Parallel data

For the simulation experiments, we use the Europarl v7 corpus, with English as the source language and each of eight languages as the target language. There are an average 1.85 million parallel sentences for each of the language pairs. For the real-world experiments, parallel data is smaller and generally of a lower quality. For Malagasy, we use a web-sourced collection of parallel texts.[4] The parallel data of Malagasy has 100k sentences and 1,231k tokens. For Kinyarwanda, we obtained

---

[4]http://www.cs.cmu.edu/ ark/global-voices/

|  | da | nl | de | el | it | pt | es | sv | Average |
|---|---|---|---|---|---|---|---|---|---|
| BiLSTM Annotated | 89.3 | 87.4 | 89.5 | 88.1 | 85.9 | 89.5 | 90.6 | 84.7 | 88.1 |
| BiLSTM Projected | 64.4 | 81.9 | 81.3 | 78.9 | 80.1 | 81.9 | 81.2 | 74.9 | 78.0 |
| BiLSTM Ann+Proj | 85.4 | 88.9 | 90.2 | 84.2 | 86.1 | 88.2 | 91.3 | 83.6 | 87.2 |
| MaxEnt Supervised | 90.1 | 84.6 | 89.6 | 88.2 | 81.4 | 87.6 | 88.9 | 85.4 | 86.9 |
| Duong et al. | 92.1 | 91.1 | 92.5 | 92.1 | 89.9 | 92.5 | 91.6 | 88.7 | 91.3 |
| BiLSTM+noise layer | 92.3 | 91.7 | 92.5 | 92.8 | 90.2 | 92.9 | 92.4 | 89.1 | 91.7 |

Table 1: The POS tagging accuracy for various models in 8 languages: Danish (da), Dutch (nl), German (de), Greek (el), Italian (it), Portuguese (pt), Spanish (es) Swedish (sv). The top results of the second part are taken from (Duong et al., 2014a), evaluated on the same data split.

parallel texts from ARL MURI project.[5] The parallel data of Kinyarwanda has 11k sentences and 52k tokens.

### 4.1.2 POS projection

We use Giza++ to induce word alignments on the parallel data (Och and Ney, 2003), using IBM model 3. Following prior work (Duong et al., 2014b), we retain only one-to-one alignments. Using all alignments, i.e., many-to-one and one-to-many, would result in many more POS-tagged tokens, but also bring considerable additional noise. For example, the English laws (NNS) aligned to French les (DT) lois (NNS) would end up incorrectly tagging the French determiner les as a noun (NNS). We use the Stanford POS tagger (Toutanova et al., 2003) to tag the English side of the parallel data and then project the label to the target side. As we show in the following section, and as confirmed in many studies (Täckström et al., 2013; Das and Petrov, 2011) the directly projected labels are very noisy and it is unwise to use the tags directly. We further filter the noise using the approach of Yarowsky et al. (2001) which selects sentences with the highest sentence alignment scores from IBM model 3. For 8 languages in Europarl corpus, we collect 200k sentences for each language. For the low-resource languages, we use the whole parallel data because of limited bilingual text.

### 4.1.3 Annotated data

Gold annotated data is expensive and difficult to obtain, and thus we assume that only a small annotated dataset is available. For the simulation experiments, we use the CoNLL data (Buchholz and Marsi, 2006) as annotated data for eight languages. To simulate the low-resource setting, we

take the first 1,000 tagged tokens for training and the remaining data is split equally between development and testing sets, following Duong et al. (2014a). For the real-world experiments, we use the Malagasy and Kinyarwanda data from Garrette and Baldridge (2013), who showed that a small annotated dataset could be collected very cheaply, requiring less than 2 hours of non-expert time to tag 1000 tokens. This constitutes a reasonable demand for cheap portability to other low-resource languages. We use the datasets from Garrette and Baldridge (2013), constituting training sets of 383 sentences and 5,294 tokens in Malagasy and 196 sentences and 4,882 tokens for Kinyarwanda. There are similar sized datasets used for testing.

### 4.2 Setup and baselines

We compare our algorithm with several baselines, including the state-of-the-art algorithm from Duong et al. (2014a), a two-output maxent model, their reported baseline method of a supervised maximum entropy model trained on the annotated data, and our BiLSTM POS tagger trained directly from the annotated and/or projected data (denoted BiLSTM Annotated, Projected and Ann+Proj for the model trained on union of the two datasets). For the real low-resource languages, we also compare our algorithm with Garrette et al. (Garrette and Baldridge, 2013), which showed good results on the two low-resource languages. Our implementation is based on clab/cnn. [6] In all cases, the BiLSTM models use 128 dimensional word embeddings and 128 dimensional hidden layers. We set learning rate as 1.0 and use stochastic gradient descent model to learn the parameters.

We evaluate all algorithms on the gold testing sets, evaluating in terms of tagging accuracy. Following standard practice in POS tagging, we re-

---

[5]The dataset was provided directly by Noah Smith.

[6]https://github.com/clab/cnn

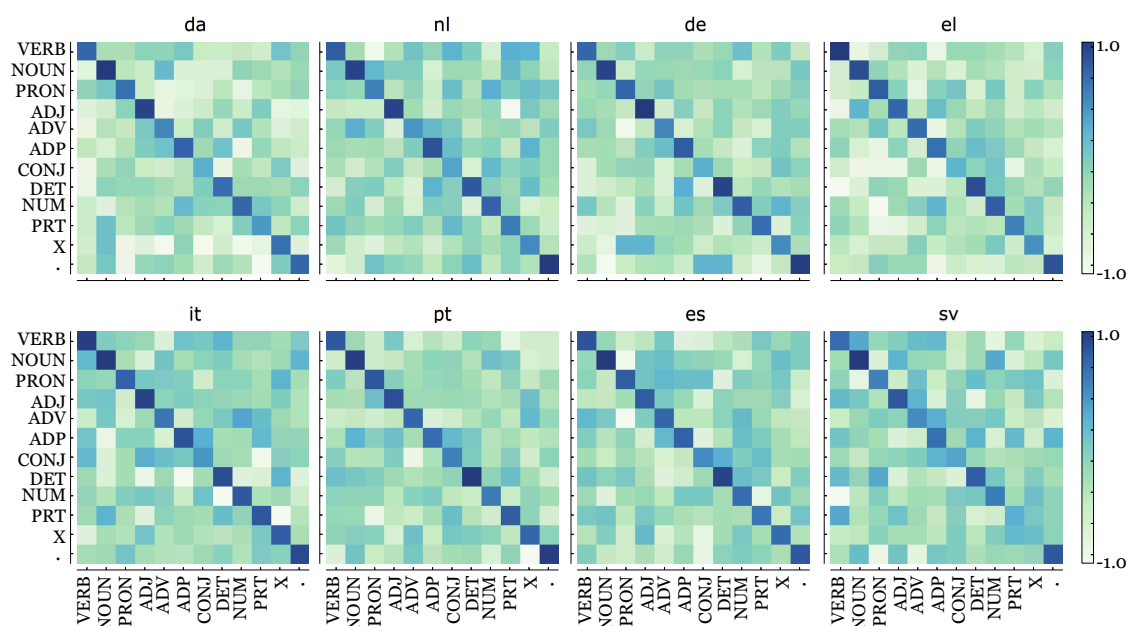

Figure 3: Noise transformation matrix $A$ between POS tags and noised (projection) outputs, shown as columns and rows, respectively, for the 8 languages.

port results using per-token accuracy, i.e., the fraction of predicted tags that exactly match the gold standard tags. Note that for all our experiments, we work with the universal POS tags and accordingly accuracy is measured against the gold tags after automatic mapping into the universal tagset.

### 4.3 Results

First, we present the results in the eight simulation languages in Table 1. For most of languages, our method is better than Duong et al. (2014a) and the three naive BiLSTM baselines. Directly training on projected data hurts the performance, e.g., compare BiLSTM Projected and BiLSTM Ann+Proj. BiLSTM Annotated mostly outperforms MaxEnt Supervised, but both methods are worse than Duong et al. and our BiLSTM+noise layer, which both use the projected data more effectively. The result shows the noise layer provides a better use of the noisy projected data, improving the POS tagging accuracy.

We show the noise layer for the different languages in Figures 3. The blue (dark) cells in the grids denote values that are most highly weighted. Note the strong diagonal, showing that the tags are mostly trusted, although there is also evidence of considerable noise. The worst case is in Swedish (sv) with many weak values on the diagonal. In

this case, PRT and X appear to be confused for one another. The white grids are also important, showing tag combinations that the model learns to ignore, such as NUM vs DET in Italian (it) and NOUN vs PRON in Spanish (es) and Swedish (sv). It shows these types are not confused. The tokens that are NUM in Italian (it) are seldom projected as DET. Overall, the level of noise looks to be modest, which might not come as a surprise given the large clean parallel corpus for learning word alignments.

Now we present the results for two low-resource languages, Malagasy and Kinyarwanda which both have much smaller parallel corpora. The results in Table 2 show that our method works better than all others in both languages, with a similar pattern of results as for the eight simulation languages. Note that our method outperforms the state of the art on both languages (Duong et al., 2014a; Garrette and Baldridge, 2013).

To better understand the effect of the noise layer, we present the learned transformation matrices $A$ in Figure 4. Note the strong diagonal for Malagasy in Figure 4, showing that each tag is most likely to map to itself, however there are also many high magnitude off diagonal elements. For instance nouns map to not just noun, but also adjective and number, but never pronoun (which

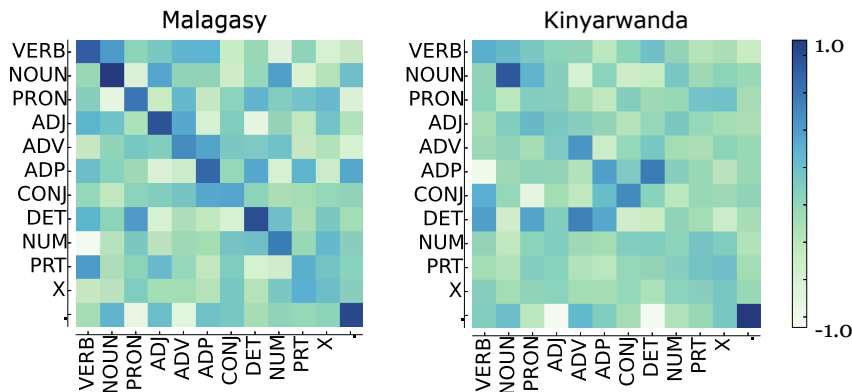

Figure 4: Noise transformation matrix $A$ between POS tags and noised (projection) outputs, shown as columns and rows, respectively, for the two low-resource languages.

| Model | Accuracy | |
| --- | --- | --- |
| | Malagasy | Kinyarwanda |
| BiLSTM Annotated | 81.5 | 76.9 |
| BiLSTM Projected | 67.2 | 61.9 |
| BiLSTM Ann+Proj | 78.6 | 73.2 |
| MaxEnt Supervised | 80.0 | 76.4 |
| Duong et al. | 85.3 | 78.3 |
| BiLSTM+noise layer | 86.3 | 82.5 |
| Garrette et al. | 81.2 | 81.9 |

Table 2: The POS tagging accuracy for various models in Malagasy and Kinyarwanda. The top results of the second part are taken from (Duong et al., 2014a), evaluated on the same data split.

are presumably well aligned.) Comparing results of Malagasy and Kinyarwanda in Figure 4, we can see the amount of noise is much greater in Kinyarwanda. This tallies with the performance results, in which we get stronger results and a greater improvement on Malagasy from using projection data, where we had more parallel data.

## 5   Conclusion

In this paper, we presented a technique for exploiting noisy cross-lingual projected annotations alongside a small amount of annotation data, in the context of POS tagging. To utilize both sources of data, we proposed a new model based on a bidirectional long short term memory recurrent neural network, with a layer for explicitly handling noisy projection labels. In two real low-resource languages, our methods outperform other algorithms. Our technique is general, and is likely to prove useful for exploiting other noisy annotations such as distant supervision and crowd-sources annotations, and with other modelling approaches.

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
