# Peer review of "Learning when to trust distant supervision: An application to low-resource POS tagging using cross-lingual projection"

_CoNLL 2016 — decision unknown_

[Official Review · Reviewer 1 · rating 4 · confidence 4]
soundness 4 · originality 3 · clarity 4 · impact 3 · substance 4 · appropriateness 5 · meaningful comparison 4 · replicability 3 · presentation format Poster

I reviewed this paper earlier, when it was an ACL 2016 short paper draft. At
that point, it had a flaw in the experiment setup, which is now corrected.

Since back then I suggested I'd be willing to accept the draft for another *ACL
event provided that the flaw is corrected, I now see no obstacles in doing so.

Another reviewer did point out that the setup of the paper is somewhat
artificial if we focus on real low-resource languages, relating to the costs of
*finding* vs. *paying* the annotators. I believe this should be exposed in the
writeup not to oversell the method.

There are relevant lines of work in annotation projection for extremely
low-resource languages, e.g., Johannsen et al. (2016, ACL) and Agic et al.
(2015, ACL). It would be nice to reflect on those in the related work
discussion for completeness.

In summary, I think this is a nice contribution, and I vote accept.

It should be indicated whether the data is made available. I evaluate those
parts in good faith now, presuming public availability of research.

[Official Review · Reviewer 2 · rating 4 · confidence 4]
soundness 5 · originality 4 · clarity 4 · impact 4 · substance 4 · appropriateness 5 · meaningful comparison 4 · replicability 3 · presentation format Oral Presentation

The paper describes a modification to the output layer of recurrent neural
network models which enables learning the model parameters from both gold and
projected annotations in a low-resource language. The traditional softmax
output layer which defines a distribution over possible labels is further
multiplied by a fully connected layer which models the noise generation
process, resulting in another output layer representing the distribution over
noisy labels. 

Overall, this is a strong submission. The proposed method is apt, simple and
elegant. The paper reports good results on POS tagging for eight simulated
low-resource languages and two truly low-resource languages, making use of a
small set of gold annotations and a large set of cross-lingually projected
annotations for training. The method is modular enough that researchers working
on different NLP problems in low-resource scenarios are likely to use it.

From a practical standpoint, the experimental setup is unusual. While I can
think of some circumstances where one needs to build a POS tagger with as
little as 1000 token annotations (e.g., evaluations in some DARPA-sponsored
research projects), it is fairly rare. A better empirical validation of the
proposed method would have been to plot the tagging accuracy of the proposed
method (and baselines) while varying the size of gold annotations. This plot
would help answer questions such as: Does it hurt the performance on a target
language if we use this method while having plenty of gold annotations? What is
the amount of gold annotations, approximately, below which this method is
beneficial? Does the answer depend on the target language?

Beyond cross-lingual projections, noisy labels could potentially be obtained
from other sources (e.g., crowd sourcing) and in different tag sets than gold
annotations. Although the additional potential impact is exciting, the paper
only shows results with cross-lingual projections with the same tag set. 

It is surprising that the proposed training objective gives equal weights to
gold vs. noisy labels. Since the setup assumes the availability of a small gold
annotated corpus, it would have been informative to report whether it is
beneficial to tune the contribution of the two terms in the objective function.


In line 357, the paper describes the projected data as pairs of word tokens
(x_t) and their vector representations \tilde{y}, but does not explicitly
mention what the vector representation looks like (e.g., a distribution over
cross-lingually projected POS tags for this word type). A natural question to
ask here is whether the approach still works if we construct \tilde{y} using
the projected POS tags at the token level (rather than aggregating all
predictions for the same word type). Also, since only one-to-one word
alignments are preserved, it is not clear how to construct \tilde{y} for words
which are never aligned.

Line 267, replace one of the two closing brackets with an opening bracket.